# Unconventional Peptide Presentation by Classical MHC Class I and Implications for T and NK Cell Activation

**DOI:** 10.3390/ijms21207561

**Published:** 2020-10-13

**Authors:** Dirk M. Zajonc

**Affiliations:** Cancer Immunology Discovery, Oncology R&D Group, WRDM, Pfizer Inc. 10555 Science Center, San Diego, CA 92121, USA; Dirk.Zajonc@pfizer.com

**Keywords:** MHC, antigen presentation, T cell activation, NK cells, killer immunoglobulin receptor

## Abstract

T cell-mediated immune recognition of peptides is initiated upon binding of the antigen receptor on T cells (TCR) to the peptide-MHC complex. TCRs are typically restricted by a particular MHC allele, while polymorphism within the MHC molecule can affect the spectrum of peptides that are bound and presented to the TCR. Classical MHC Class I molecules have a confined binding groove that restricts the length of the presented peptides to typically 8–11 amino acids. Both N- and C-termini of the peptide are bound within binding pockets, allowing the TCR to dock in a diagonal orientation above the MHC-peptide complex. Longer peptides have been observed to bind either in a bulged or zig-zag orientation within the binding groove. More recently, unconventional peptide presentation has been reported for different MHC I molecules. Here, either N- or C-terminal amino acid additions to conventionally presented peptides induced a structural change either within the MHC I molecule that opened the confined binding groove or within the peptide itself, allowing the peptide ends to protrude into the solvent. Since both TCRs on T cells and killer immunoglobulin receptors on Natural Killer (NK) cells contact the MHC I molecule above or at the periphery of the peptide binding groove, unconventionally presented peptides could modulate both T cell and NK cell responses. We will highlight recent advances in our understanding of the functional consequences of unconventional peptide presentation in cellular immunity.

## 1. Antigen Presentation by MHC I and II and T Cell Activation

Major histocompatibility (MHC) Class I and II molecules are expressed on professional antigen-presenting cells (APCs), including dendritic cells (DCs) and B cells, and present peptides to the antigen receptor (TCR) on the surface of CD4+ and CD8+ T cells [1,2,3]. The main function of MHC I and II molecules is to screen the peptide pool of APCs for viral or altered self-peptides to detect infection or cancer, while the uncontrolled response to unaltered self-peptides can induce autoimmunity [4,5]. Peptides are generally processed from proteins in the proteasome, while the N-terminus is further trimmed in the ER by specific amino-peptidases (ERAP1 and 2). Peptide processing is not the focus of this review; however, several excellent review articles cover this topic [6,7,8,9].

While MHC II presents mostly exogenous foreign peptides, MHC I presents endogenous peptides, such as self or viral peptides. Since certain viruses evolved the ability to directly interfere with antigen processing and presentation of CD8+ DCs, cross-presentation of viral proteins by a non-infected APC is one mechanism to provide immunity against viral infection [10,11].

The TCR generally binds over the central portion of the presented peptide in a diagonal binding orientation, thereby “reading out” the peptide sequence through direct interaction with accessible peptide side chains (Figure 1). TCR binding to the peptide MHC (pMHC) complex initiates the assembly of the immunological synapse between the antigen-presenting cell and the T cell by recruiting co-stimulatory molecules on both cells and clustering of the TCR [12,13,14,15,16]. This initiates a signaling cascade within the T cell, leading to transcription of genes necessary for both T cell activation and proliferation [17].

While the general mechanism of CD8+ and CD4+ T cells is similar, the molecular details in the presentation of the MHC I and MHC II bound peptides are markedly different. MHC class II molecules have an open antigen-binding groove and can bind longer peptides (>15 amino acids) that protrude with both the N- and C-terminus outside of the lateral binding groove (Figure 2). In contrast, MHC I has a closed binding groove where the conserved residues Trp167 and Tyr84 line the A and F pocket, respectively [22]. As a consequence, the peptide repertoire that any given MHC I molecule can efficiently bind is restricted in length. The majority of MHC I peptides are 8–10 amino acids long (9-mers are preferred) and longer peptides either bind in a zig-zag orientation [23] inside the groove or bulge out of the middle of the groove [1,2,22,24,25]. Allele-specific binding pockets favor certain anchor residues (e.g., P2 and P9) and provide peptide ligand specificity for polymorphic MHC I molecules of all classical haplotypes (HLA-A, -B, and -C). The preference of these pockets for certain amino acids allowed the development of algorithms to predict peptide binding to both MHC class I and II [26,27,28,29]. Peptide binding predictions are an important tool to identify the HLA allele that can present any given peptide and thus indicate whether it can potentially mount a T cell response. Identifying peptides, especially neo-antigens, through next-generation sequencing (NGS) of patient-derived tumors together with MHC binding predictions is the starting point for personalized T cell-based immunotherapies, such as adoptive T cell therapy [30,31,32,33,34].

## 2. NK Cell Recognition of MHC I: A Check and Balance System

However, T cells are not the only immune cells that can bind to and recognize MHC molecules. NK cells contribute about 10% of all circulating lymphocytes and are constantly patrolling [35]. They are very effective in destroying virally infected cells and tumor cells. Unlike T cells, receptors on NK cells that engage MHC-like molecules can be less specific for the presented peptide antigen and predominantly bind MHC I itself. When the peptide participates in the binding interaction with the NK receptor, it often modulates the binding affinity of the interaction [36]. Inhibitory receptors on NK cells (KIRs, CD94/NKG2A) constantly screen for MHC I expression on APCs in an effort to detect downregulation of MHC I by viruses, which have evolved mechanisms to evade T cell recognition and killing [37,38,39,40]. Activating receptors, such as NKG2D, on the other hand, engage non-classical MHC I and MHC I-like molecules that are upregulated upon cellular stress or infection, such as retinoic acid early inducible gene 1 (RAE-1), Histocompatibility 60 (H-60), MHC class I chain-related gene A (MICA), B (MICB), and UL16 binding proteins (ULBPs) [41]. MICA is an MHC I-like molecule that does not associate with β2-microbgobulin (β2M) and does not bind peptides. In contrast, HLA-E presents leader peptides of classical MHC I molecules as a means of detecting their cell-surface expression [42,43]. HLA-E can both engage inhibitory (NKG2A/CD94) and stimulatory (NKG2C/CD94) receptors and the presented peptide is predominantly contacted by the CD94 subunit, while both NKG2A and C have overlapping and distinct binding sites on HLA-E, resulting in a a 6-fold higher binding affinity of the inhibitory interaction with NKG2A/CD94 [36,44,45].

NK cell receptor binding to MHC I is less restricted compared to TCRs and can occur at the peptide–MHC I interface or below the peptide binding platform (Figure 1). The binding of the NKG2A/CD94 heterodimer to HLA-E roughly parallels the binding mode of the NKG2D homodimer to MICA. In a healthy cell, interactions between inhibitory NK receptors dominate, leading to the inhibition or tolerization of NK cells against the self. Upon infection, the balance between positive (activating receptors) and negative (inhibitory receptors) signals is changed in favor of NK cell activation, due to the upregulation NK cell-activating ligands, leading to NK cell activation and target cell killing. In summary, NK cells form a check and balance system by monitoring the expression of MHC molecules on cells.

## 3. Unconventional Peptide Presentation

### 3.1. Small Molecule Modulation of Peptide Presentation

Small molecules can directly alter the presentation of canonical MHC I-binding peptides and alter their recognition by T cells. The most prominent example is abacavir, a nucleoside analog that inhibits the reverse transcriptase of HIV and is used to treat HIV infection [46]. Although abacavir is generally well-tolerated, hypersensitivity reactions to abacavir developed in a small percentage of treated patients [46]. Genetic testing established a correlation between the HLA genotype and hypersensitivity [47,48]. It has since been well documented that patients with the HLA-B*57:01 allele are at risk of developing hypersensitivity reactions, and structural data deciphered the molecular mechanism leading to an altered presentation of self-peptides and the induction of an aberrant T cell response [49,50]. Abacavir binds at the bottom of the F pocket of HLA-B*57:01, thereby directly changing the shape of the F pocket (Figure 3). Specifically, abacavir alters the size of the F pocket, and peptides with shorter anchor residues at P9 (Ile and Leu versus Trp) are now favored to bind. A tryptophane residue would sterically clash with the abacavir molecule, leading to an altered pool of peptides that can be bound (Figure 3). Since the immune system has not encountered these altered pMHC I complexes, no central tolerance has been established, which could result in a cytotoxic T cell response as the trigger of the hypersensitivity reaction.

One could argue that small molecule-induced alteration of the shape of the peptide binding groove of MHC I alleles does not directly affect the binding orientation of any given peptide but rather alters the pool of peptides that can be bound and presented altogether. Nevertheless, the functional ramifications of the small molecule binding could be considered an “unconventional” peptide presentation because these peptides are not be presented in the absence of abacavir or other potential small molecule-binding groove modifiers, such as carbamazepine (CBZ). CBZ is an anti-epileptic drug but severe hypersensitivity reactions have been associated with HLA-B*15:02, while milder symptoms occurred in patients carrying HLA-A*31:01 [51,52,53]. In addition, HLA-B*15:02 patients treated with CBZ had a restricted TCR β-chain usage [54], suggesting an alteration of the presented peptide pool. However, HLA-A*31:01 differs substantially in sequence compared to HLA-B*15:02 and both alleles have different preferred anchors in the F pocket at P9 (non-polar aromatic for B*15:02 and arginine for A*31:01). In addition, the onset of the hypersensitivity reaction was delayed in HLA-B*15:02 patients, compared to those carrying HLA-A*31:01, suggesting a different mechanism of action in both populations. Indeed, it has since been shown that CBZ is degraded to its main metabolite CBZ-10,11-epoxide (EPX) via cytochrome P450. Similar to abacavir, EPX can bind in the F pocket of HLA-B*15:02, thereby altering the peptide pool [53,55]. Less data is available on the molecular basis of the milder and quicker adverse reaction in CBZ-treated patients carrying the HLA-A*31:01 allele, but it has been shown that CBZ, rather than its metabolite EPX, can directly bind and alter the amino acid preference of high-affinity binding peptides while low-affinity self-peptides might still be co-presented [53].

### 3.2. Unconventional Presentation of N- and C-Terminally Extended Peptides

#### 3.2.1. Presentation of C-Terminally Extending Peptides

More direct changes in peptide presentation can be observed using conventional MHC I presented peptides that are extended by a few additional amino acids at either the N- or C-termini. The first structural hints stemmed from the Wiley lab, who first reported the structure of a calreticulin 10-mer peptide bound to HLA-A*02:01 [56]. The peptide terminated with a glycine, which protruded with its carboxy terminus from the F pocket. Since glycine is the smallest amino acid, it did not induce any major structural changes that would open the F pocket. Instead, a slight rotation along the long axis within Tyr84 was apparently sufficient to accommodate most of its main chain [56]. Similarly, a study from the Deisenhofer lab suggested the C-terminal amino acid protrusion of a 9-mer peptide from the F pocket of H2-M^3^ [57]. However, the peptide C-termini was disordered and no structural change was observed that opened the F pocket. A few years later, using both MHC I-peptide refolding and computational modeling, the Nathenson lab suggested that C-terminally but not N-terminally extending peptides can open the F pocket of mouse H-2K^b^ to allow the C-terminal extension to extend from the peptide binding groove and into the solvent [58]. In this case, the C-terminal extensions contained hydrophobic amino acids. More recently, the Cole group demonstrated the opening of the F pocket via crystallography using the mouse H-2K^d^ allele and an insulin-derived 10-mer peptide that contained a C-terminal phenylalanine reside [59]. The authors observed that Tyr84, which forms a lateral wall that closes the F pocket, swung out to open the peptide binding groove. Surprisingly, the authors observed the altered position of T84 in two out of the three complexes contained within the asymmetric unit of the crystal and none of the peptides demonstrated clear electron density except for a few N-terminal residues. As such, a structural explanation of why Y84 swung out was not obvious. Nevertheless, these studies suggested that the presence of terminal hydrophobic amino acids in longer peptides can open the F pocket of some murine MHC I alleles. However, hydrophobic or aromatic residues are often found as the last anchor residue in the P9 (PΩ) position of conventional peptides and increasing the peptide length up to 15-mers generally leads to a bulging of these peptides in the center of the groove whilst not affecting the confinement within the A or F pockets. Therefore, it is unclear what factors control the opening of the F pocket and how we can predict it and potentially exploit that mechanism therapeutically.

Work involving several labs, including Hildebrand, Peters, and our own, further identified and characterized peptides that were eluted from *Toxoplasma gondii*-infected human THP-1 cells expressing HLA-A*02:01 [60]. Bioinformatic analysis revealed that the *T. gondii* peptides were significantly longer than the host peptides, and contained a canonical N-terminal binding core but carried C-terminal amino acid extensions of 3–12 residues. No conserved amino acid signature was found in the C-terminal extensions; however, all peptides contained either positively or negatively charged (or both) amino acids. Binding algorithms failed to predict the binding of these longer peptides to HLA-A*02:01, likely because no anchor residue was found in the C-terminal extension. However, both core and extended peptides refolded similarly well with HLA-A*02:01 and also bound with high affinity to MHC I [60]. Compared to the core peptide, the C-terminally extending peptides formed stable complexes with MHC I, based on thermal denaturation fluorimetry data. The crystal structures of various conventional and C-terminally extending peptides bound to HLA-A*02:01 were then determined [60,61]. We noticed that while the core peptides bound as expected with the N- and C-terminal amino acid tugged into the MHC binding groove, the C-terminally extending peptides induced a structural change around the F pocket of HLA-A*02:01. This structural change involved the residues Tyr84 and Lys146, which form the lateral wall (Tyr84) and the lid (Lys146) and together close the F pocket (Figure 4). While the 11-mer peptide F11V (FVLELEPEWTV) bound in a zig-zag orientation, the addition of a lysing in the 12-mer peptide F12K (FVLELEPEWTV-**K**) induced the “Tyr84 swing” mechanism to open the F pocket (Figure 4B). The opening of the F pocket resembled the structural change that the Cole lab observed for the mouse allele; however, in their study, it was a result of a hydrophobic amino acid extension [58]. Most extending peptides that we identified upon *T. gondii* infection, however, contained a negatively charged residue, either following the core peptide directly or within a few residues. Longer peptides with negatively charged additions did not alter the orientation of Tyr84; however, Lys146 moved up into the solvent (“Lys146 lift”) (Figure 4D). The Tyr84 swing appears to require greater flexibility of the protein backbone, and we hypothesize that amino acid 80 (here Thr80) was involved in the structural change. Thr80 adopted a different rotamer and appeared to initiate a slight shift in the protein backbone that continued well beyond Tyr84 and likely allowed Tyr84 to swing out. In summary, we identified two mechanisms of opening the F pocket of HLA-A*02:01, depending on the nature of the first charged amino acid that follows the peptide binding core. Adding a hydrophobic amino acid, such as that found in the peptide Y10L (YLSPIASPLL), did not lead to the opening of the F pocket but resulted in a more compact peptide binding compared to the core peptide YLSPIASPL, where the terminal leucine residue was used as the anchor residue in the F pocket (P10 instead of P9) [61]. Addition of the negatively charged C-terminal amino acids, however, opened the F pocket and resulted in more relaxed binding of the 9-mer core peptide Y9L, with leucine at P10 now exiting the F pocket into the solvent (Figure 4D). Therefore, the molecular details that lead to the opening of the F pocket in HLA-A*02:01 differed from the earlier study by Dr. Nathenson and Dr. Cole for mouse MHC I alleles [58,59]. In agreement with the murine studies, the Gfeller lab recently characterized the binding of C-terminally extended peptides for other HLA-A alleles (A*03:01 and A*68:01) that used a hydrophobic amino acid extension to open the F pocket via the Tyr84 swing mechanism [62]. Since these alleles have a positively charged anchor residue at P9, addition of a hydrophobic residue was necessary for the F pocket opening, while alleles that have hydrophobic anchor residues at P9 (e.g., A*02:01) required a charged residue to break the F pocket confinement (Figure 4C). In summary, the possibility that peptides can protrude from the F pocket into the solvent has been reported for both mouse and human MHC I molecules. Since the residues Tyr84 and Lys146 are conserved across all HLA-A, -B, and -C haplotypes, we speculate that opening of the gate of the F pocket is a universal mechanism to allow C-terminally extending peptides to protrude into the solvent and we demonstrated the binding of designed C-terminally extending peptides also for certain HLA-B and HLA-C alleles (US patent application 16/348,839).

#### 3.2.2. Presentation of N-Terminally Extending Peptides

One could argue that peptides with C-terminal extensions could more easily break confinement and project out of the F pocket, since the carboxyl group of their last anchor residue in P9 already faces up toward the solvent. In contrast, the N-terminal amino group of conventionally bound peptides faces toward the bottom of the A pocket and it was not conceivable how peptides with N-terminal extensions could be bound and presented. Surprisingly, a recent study demonstrated that HLA-B*57:01 can bind N-terminally extending peptides by projecting their N-termini out of the A pocket [63]. In this study, Rossjohn, Purcell, Vivian, and colleagues identified that the repertoire of the HLA-B*57:01 presented HIV-1 Gag epitope contained 20% of peptides that had such N-terminal additions. Among those, the immunodominant epitope TW10 (**TST**LQEQIGW) was bound with the N-terminal amino acids extending out of the A pocket of HLA-B*57:01 [63]. The N-terminal additions contained the same amino acids at P-1 and P1 compared to the anchor residue at P2 (Ser, Thr, and Ala). The authors first speculated that this could have led to the N-terminal additions forming alternate anchors, while the middle portion of the peptides bulged out of the groove. Instead, the authors observed that these peptides bound in an orientation in which the serine residue, at P1 of the N-terminally extending peptides, bound in the A pocket where usually the N-terminal amide of conventionally bound peptides was accommodated. However, in N-terminally extending peptides, the orientation of the serine was altered so that the N-terminus faced upward, allowing the N-terminal residue at P-1 to protrude out of the A pocket. Interestingly, this binding mode is recognized by CD8+ T cells and it is instead the viral escape mutation T3N, where the threonine anchor residue at P2 of TW10 is replaced by asparagine, that resulted in a complete loss of T cell recognition. Maybe more surprisingly, the escape mutant peptide bound in the conventional binding mode, with both N- and C-termini contained in the peptide binding groove, while the middle portion bulged out of the groove, thereby evading T cell recognition likely via steric clashes with the TCR.

In addition, peptides that protrude with their N-terminus out of the A pocket have also been reported for the non-classical MHC I allele HLA-F. HLA-F can exist as an open conformer in the absence of bound peptides, but it can also bind peptides of a length that is more similar to MHC II molecules [64]. Several residues that are conserved across MHC I alleles have been substituted in HLA-F, thereby blocking off the A pocket and forcing the peptide to extend upward and out of the groove. As such, the structural uniqueness of HLA-F and not a specific amino acid signature within the presented peptide is the result of the unconventional presentation of HLA-F bound peptides [64].

In summary, the unconventional presentation of terminally extending peptides has been observed across different MHC I alleles and is either dictated by allele-specific amino acid additions (HLA-A, and -B) or structural alterations of the peptide binding groove (HLA-F). Nevertheless, at least for N-terminally extending peptides, TCRs are still able to engage the pMHC complex, resulting in the activation of cytotoxic T cells.

## 4. Killer Immunoglobulin Receptors (KIRs) in Infection and Cancer

TCRs expressed on CD8+ T cells are not the only receptors that recognize MHC I molecules. Killer immunoglobulin receptors (KIRs) are a family of receptors expressed on NK cells, as well as on some T cell subsets, and bind to MHC I molecules, preferentially the HLA-B and -C haplotypes but also some HLA-A alleles [65] and the open conformer of HLA-F [64]. KIRs are composed of either two (KIR2D) or three (KIR3D) immunoglobulin (Ig) domains and express either a short (S) or long (L) cytoplasmic tail. KIRS are activating receptors, since they can associate with DAP12, while KIRL are inhibitory receptors that encode ITIM motives in their cytoplasmic domains [66,67]. Structural data revealed that inhibitory KIRs bind above the F pocket of HLA-B [21] and HLA-C [68,69] molecules. Similar to its MHC I ligands, KIR receptors have multiple alleles and are highly polymorphic in sequence (https://www.ebi.ac.uk/ipd/kir/). Ligands for inhibitory KIRs are well defined, while ligands for KIRSs are largely unknown. In viral infections, particular combinations of NK-activating receptors and their ligands are protective. For example, the presence of KIR3DS1 and its putative ligand HLA-Bw4 (I80) was identified as a key factor in preventing HIV infection from leading to full-blown AIDS [70,71,72]. In hepatitis C infection, KIR2DL3 homozygosity and HLA-C1 homozygosity are beneficial in both early eradication of infection and response to standard treatment (type I IFN + ribavirin administration) [73,74]. Homozygosity of KIR2DL3 and HLA-C1 alleles has been reported to lead to lower levels of NK inhibition than other pairs of KIR ligand combinations [75,76], suggesting that this underlies the enhanced response to hepatitis C.

NK cells are versatile killers, and their activation has clinical relevance for various diseases [77]. Tumor infiltration of NK cells in humans may be associated with a better prognosis in various solid tumors, such as squamous cell, lung, gastric, and colorectal carcinoma [78]. Especially, tumors that downregulate classical MHC I expression to evade TCR mediated recognition, or upregulate non-classical MHC I activating ligands (altered-self recognition) are ideal targets for NK cells [79,80]. NK cells were also shown to be effective in the eradication of established hematological diseases, such as acute myeloid leukemia (AML), multiple myeloid lymphoma (MML), and chronic myeloid lymphoma (CML). Recently, the administration of the anti-inhibitory KIR antibody IPH2101 to autologous multiple myeloma (MM) cells enhanced NK cell cytotoxicity against the tumor cell but not normal cells by blocking the interaction of KIR2DL1–3 with HLA-C [81,82]. These data demonstrate that disrupting a single inhibitory KIR interaction with its MHC I ligand can lead to NK cell activation and tumor killing, while NK cell tolerance against normal cells in not impaired.

## 5. Outlook and Future Perspectives

While N-terminally extending peptides presented by HLA-B*57:01 have been shown to modulate the binding affinity of KIR3DL1 by altering the peptide interactions with the receptor [21], the binding interaction between the receptor and the HLA molecule is not impacted to an appreciable extend. In addition, alanine scanning mutagenesis of HLA-B*57:01 identified that the F pocket residue Lys146 (involved in the lysine lift mechanism) is absolutely required for KIR3DL1 binding [21]. Therefore, we hypothesize that C-terminally extending peptides disrupt the major binding site for KIRs and will fully abrogate KIR binding, thereby modulating NK cell activity. Furthermore, since Tyr84 and Lys146 are conserved across all HLA I haplotypes, we will likely identify many more HLA I alleles that can open the F pocket and present C-terminally extending peptides. We hypothesize that the interaction between inhibitory KIRs and the corresponding HLA allele will be disrupted by the structural change that alters the position of both conserved residues (K145 lift and Tyr84 swing) and by the C-terminal amino acid extension, which will further provide a steric clash with the KIR receptor.

By adding charged residues to high-affinity binding and immunogenic peptides for MHC I alleles that utilize a hydrophobic anchor residue at P9 (such as HLA-A*02:01 or HLA-B*57:01), we can design peptides that could break the confinement of the F pocket and modulate NK cell activity. Conversely, MHC I alleles (such as HLA-A*03:01, A*68:01) that use positively charged anchor residues at PΩ can break confinement with the addition of a hydrophobic amino acid at the C-terminus. Whether these designed peptides could be used as a potential biotherapeutic approach to activate NK cells would depend on the efficiency of their presentation by the HLA allele. For peptide antigens that induce a CD8^+^ T cell response, only a few TCR-pMHC contacts need to be formed between the antigen-presenting cell and the T cell [83]. In contrast, overcoming inhibitory signaling by preventing the interaction between HLA and KIRDL molecules would likely require the majority of HLA molecules to present the designed peptide. Whether these peptides are transported in the ER via the TAP transporter or whether they can occupy empty HLA molecules that transiently exist on the cell surface after self-peptide dissociation is currently not known. Cell-based assays suggest that some peptides do indeed load efficiently enough in vitro to overcome inhibitory signaling, resulting in NK cell activation and target killing (US patent application 16/348,839). However, whether the same holds true in vivo is not known. It is also interesting to speculate whether we can activate both CD8+ T cells and NK cells with the same designed peptides. Since most TCRs bind in a diagonal binding orientation over the central portion of the presented peptide, the structural change at the F pocket may not affect the binding of many TCRs. This in turn would then still allow an oligoclonal CD8^+^ T cell response. Conversely, on APCs, where the peptide incorporation is sufficient enough to prevent the majority of inhibitory KIRDL interactions, we could also see an NK cell response. For a biotherapeutic approach, however, the peptides wood have to be targeted to the tumor cell to prevent systemic T cell and NK cell activation, which could lead to toxicity or even autoimmunity.

## 6. Patents

Zajonc D, Peters B, Remesh SG, Ying G, Use Of C-Terminally Extended Peptides to Disrupt Inhibitory NK Cell Receptor Interactions with MHC I. US patent application 16/348,839. 2017.

## Figures and Tables

**Figure 1 ijms-21-07561-f001:**
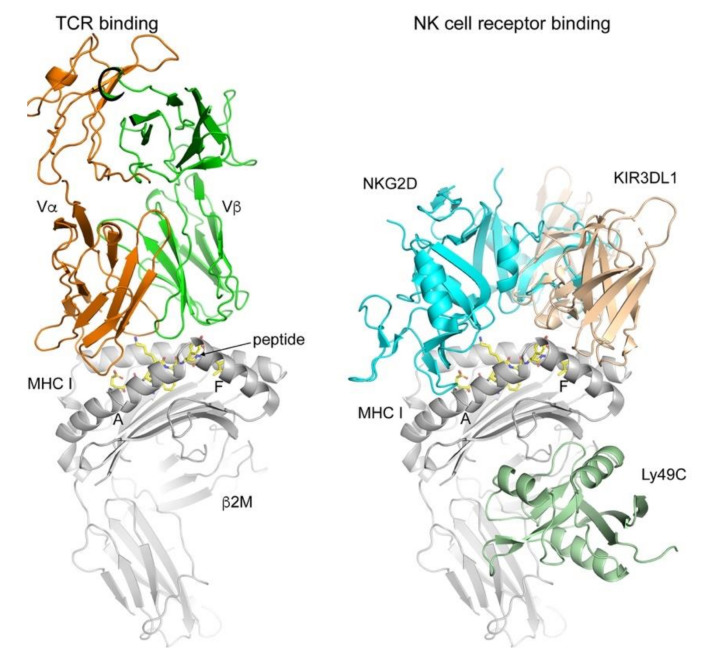
TCR and NK receptor binding to classical and non-classical MHC I molecules. The TCR binds to the pMHC I complex using a diagonal orientation while contacting both the MHC I molecule (grey) and the peptide (yellow sticks). All complexes were superimposed on the MHC I molecules and Protein Data Bank (PDB) codes used were 2CKB (2C TCR-H-2K^b^, [18]), 1HYR (NKG2D-MICA, [19]); 5J6G (Ly49C-H2-Q10, [20]); 3VH8 (KIR3DL1-HLA-B*57:01, [21]). NK receptor binding is more diverse and does not always depend on the presence of a peptide (e.g., MICA-NKG2D) or contacts the peptide (Ly49C). MHC I molecules in grey cartoon, while the receptors are colored differently. Figures prepare using PyMol (Schrodinger, LLC, New York, NY, USA).

**Figure 2 ijms-21-07561-f002:**
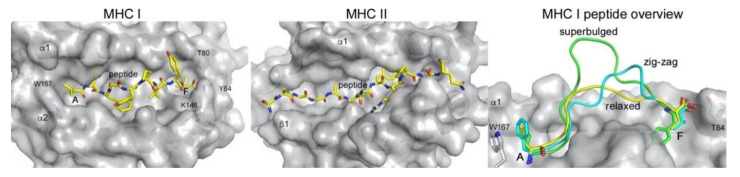
Peptide presentation by MHC class I (MHC I) and class II (MHC II). The peptide (yellow sticks) is confined between the major binding pockets A and F in MHC I (PDB 1DUZ), while it laterally extends outside the binding platform in MHC II (PDB 1K2D) due to the open groove architecture. Residues involved in closing the A and F pocket in MHC I are indicated. Right panel, MHCI can present longer peptides either bulging out in the middle of the binding groove (green, PDB 2AK4) or in a zig-zag fashion (cyan, PDB 5D9S), compared to the relaxed presentation of a 9-mer peptide (yellow, PDB 1DUZ). MHC molecules in the slightly transparent grey molecular surface, peptides as yellow sticks or as colored loops (right panel). All figures were prepared using PyMol (Schrodinger, LLC) and the indicated PDB files.

**Figure 3 ijms-21-07561-f003:**
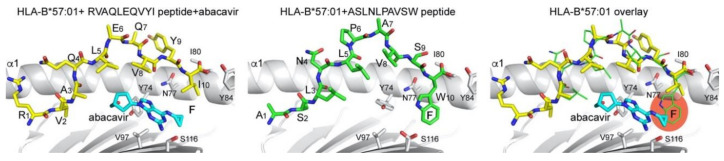
Abacavir modulation of peptide binding to HLA-B*57:01. Peptide (yellow and green sticks) and abacavir (cyan sticks) binding reveals differences in the size of the F pocket. Overlay of two different peptides that bind to HLA-B*57:01 in the absence or presence of abacavir highlight how peptides with terminal Trp anchor residues are disfavored from binding together with abacavir, due to a steric clash (right panel). PDB 3VRI was used for the abacavir-containing structure and PDB 6V2O for the structure without abacavir. Overlay was generated by superimposing both HLA-B*57:01 molecules (grey cartoon).

**Figure 4 ijms-21-07561-f004:**
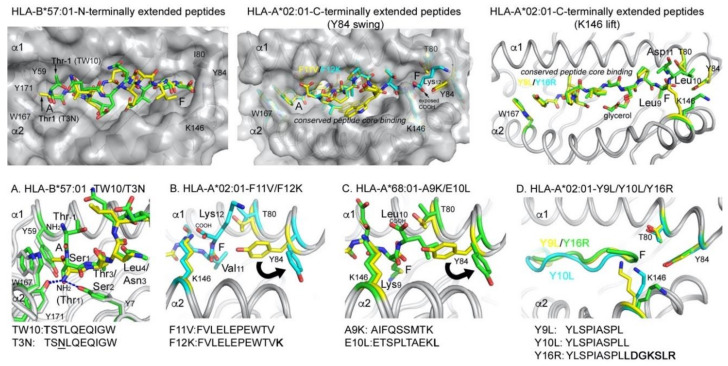
Presentation modes of N- and C-terminally extending peptides by MHC I alleles. N-terminally extending peptides undergo a structural change that allow them to protrude from the A pocket (top row, left panel), while C-terminally extending peptides instead induce a structural change in the MHC I molecule that opens the F pocket (top row, middle and right panel). (**A**) Comparison between a the conventionally presented peptide T3N (yellow, PDB 5T70) with the extending peptide TW10 (green, PDB 5T6X). (**B**) Tyr84 swing mechanism of HLA-A*02:01 with a C-terminally extended positively charged residue (peptide F12K as cyan sticks, PDB 5DDH), compared to the conventional F11V peptide (yellow sticks, PDB 5D9S). (**C**) Tyr84 swing mechanism of HLA-A*68:01 with a C-terminally extended hydrophobic residue (peptide E10L as green sticks, PDB 6EI2) compared to the conventionally presented A9K peptide (yellow, PDB 4HWZ). (**D**) Lys146 lift mechanism following negatively charged amino acid addition. Structures containing core peptides in yellow (peptide Y9L, PDB 5F9J), extending peptides in green (Y16R, PDB 5FA4) and cyan (Y10L, PDB 5FDW).

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
