# Peer review of "Unconventional Peptide Presentation by Classical MHC Class I and Implications for T and NK Cell Activation"

_ijms, 2020, doi:10.3390/ijms21207561_

Round 1

Reviewer 1 Report

Several aspects are missing, which should be included

1) the interaction between HLA-E peptide presentation and NKG2C or NKG2A.

2) the non-canonical peptide presentation by HLA-E and HLA-F

3) the effect of carbamazepine-10,11-epoxide (EPX) on peptide presentation

Please correct as it is misleading: MICA is not a non-classical MHC molecule, as (i) it is polymorphic like classical MHC, is not associated with ß2m and do not present peptides!

Please improve the following sentence: "receptors on NK cell that engage MHC like molecules are far less specific for the presented peptide antigen and mainly bind MHC I itself." Again, the presented peptide by HLA-E guides the affinity and its interaction towards the inhibitory receptor pair NKG2A/CD94 or stimulatory receptor pair NKG2C/CD94.

Author Response

We would like to thank reviewer #1 for the constructive feedback that helped us to improve the revised manuscript.

Since this article is mostly about “unconventional peptide presentation of classical MHC I molecules”, we have changed the title to reflect this.

  • We do discuss the interaction between HLA-E-peptide-NKG2A/C a bit more but only in the light of how NK cells see MHC I molecules.

  • We are not discussing the details of the peptide presentation itself (see also your question regarding HLA-E peptide presentation below). Since HLA-E has an A to F pocket architecture and presents the leader peptides of other MHC I molecules in a confined peptide binding groove, we consider the peptide presentation by HLA-E rather “conventional”.We do have included an abstract about the unconventional peptide presentation by HLA-F, as suggested.

  • We have included a chapter on the effect of carbamazepine (CBZ) and its metabolite EPX on peptide presentation. Since no structural data exists, we were not able to include that in figure 3. However, we have edited Figure 3 slightly, since we found that one amino acid was mislabeled.

  • We have also modified the sentence on MICA being an MHC I like molecule that does not bind peptides, as well as modified the sentence referring to the mostly peptide independent recognition of NK cell receptors. These sections are highlighted in yellow in the manuscript.  

Reviewer 2 Report

Summary:

The author reviews the literature related to presentation of unconventional peptides by MHC I and the potential impacts of this on activation of T cells and NK cells. Following a short overview of antigen presentation and NK cell activation, the author breaks down unconventional peptide presentation into three categories: small molecule modulation of peptide presentation, presentation of c-terminally extending peptides, and presentation of N-terminally extending peptides. This is followed by short sections describing the role of inhibitory receptors on NK and T cells, as well as a summary of future perspectives.

Specific comments:

1) There are some typographical errors in various places in the manuscript which should be corrected prior to publication. 

2) With the exception of Figure 1, there is no explanation as to how the diagrams in the figures were generated. 

3) Although a section on "Outlook and future perspectives" is included, it is rather short. A longer and more detailed explanation about the practical applications of unconventional peptide presentation for cancer immunotherapy would strengthen the review and provide additional context. Perhaps a summary or overview figure about the biological/therapeutic applications would also be helpful in this regard. 

Author Response

We would like to thank reviewer #2 for the constructive feedback that helped us to improve the revised manuscript.

  • We carefully went through the manuscript to fix several typographical and grammatical errors.

  • We have listed both the software (PyMol) and all the PDB files in each figure legend that were used to generate the figures.

  • We have expanded the “Outlook and future perspectives” as suggested but in general, the article is more tailored to the underlying mechanisms that controls T cell and NK cell activation, rather than using it as a potential therapeutic approach. We have, however, discussed some possibilities and caveats using designed peptides as biotherapeutics.

Reviewer 3 Report

pag 7 line 265 change  multiple melanoma into multiple myeloma 

pag 7 line 260 change evade inhibitory KIR signaling into evade TCR mediated recognition

Author Response

We would like to thank reviewer #3 for pointing out some minor mistakes.

We have both corrected “multiple melanoma” to “multiple myeloma” and

“evade inhibitory KIR signaling” to “evade TCR mediated recognition”.

Round 2

Reviewer 1 Report

No further comments.